



# Assessment of Hydrology Estimates from ERA5 Reanalyses in Benin (West Africa)

René Bodjrènou[a,b], Luc Ollivier Sintondji[b], Yekambessoun M'Po N'Tcha[a], Diane Germain[c], Francis Esse Azonwade[d], Ayemar Yaovi Bossa[b], Silvère Fernand Sohindji[e], Gilbert Hounnou[d],
Edid Amouzouvi[f], Arthur Freud Segnon Kpognin[g]

a) Laboratoire d'Hydrologie Appliquée (LHA), Institut National de l'Eau (INE), Benin
b) Laboratoire d'Hydraulique et de Maîtrise de l'Eau (LHME- INE), Benin
c) DG Environnement, Canada
d) Direction Générale de l'Eau (DGEau), Bénin
e) Centre of Eco-Education and Valorization of Vegetables (CEEVaL-Benin), Benin Republic
f) Direction Départementale de l'Eau, Energie et Mine (DDEEM-Mono), Bénin
g) Faculté d'Agronomie de l'Université de Parakou (FA-UP), Bénin

*Correspondence to* René Bodjrènou ([renebodjrenou@gmail.com](mailto:renebodjrenou@gmail.com))

**Abstract.** In West Africa, the validation of distributed models is limited by the quality and
availability of point station data measured in-situ. ERA5 is a climate reanalysis produced by European Centre for Medium-range Weather Forecasts (ECMWF) and suggested to overcome this constraint. This study assessed and compared over the Benin basins at spatial and monthly time scale, the quality of ERA5 and its variant ERA5-Land (namely LAND). ERA5 relies on the single-levels version with 0.25° x 0.25° resolution while LAND is the land surface version
with 0.1° x 0.1° resolution. Four variables were collected including runoff, evapotranspiration (ETR), water table depth (WTD), and soil water content (SWC). Point station data were analyzed using the correlation performance evaluators, Mean Absolute Error (m) and Relative Mean Absolute Error (r). The results showed that LAND simulates well the peaks of mean runoff. It showed the best runoff performance in terms of correlation (~0.61) compared with
ERA5 (correlation ~0.49). Both reanalysis showed high correlations (generally > 0.80) for SWC, but the correlations obtained from ETR are slightly lower (ERA5~0.58 vs. ERA5-Land~0.54). Correlations were below 0.5 on both reanalyses for WTD with slight overestimation (m=4.73m for ERA5 vs. m=3.13m for LAND). This study does not identify any reanalysis that is better than another, both spatially and monthly scale. Nevertheless, this study
indicated that the choice of reanalyses must rely on their performance and the given water cycle element. Correcting the variables of these reanalysis could also improve their performance.

**Keywords:** ERA5, ERA5-Land, soil water content, runoff, Evapotranspiration, Water Table Depth, west Africa



## 1. Introduction


Hydrological impact studies are of paramount importance in establishing effective development strategies and aimed at preserving water resources for future generations. The documentation of the impacts of current climate on water resources is growing, and sometimes their combined effects with land use (deforestation, land development, etc.) are attracting and increasing

attention from researchers (Bodjrènou and Comandan 2023a). Unfortunately, the data needed to assess the consequences of these factors remain limited in terms of quality and time. Dembélé *et al.* (2020) indicated that the irregular spatial distribution of data has hampered thorough documentation of the evolution of the water cycle in West Africa. In addition, lack of station monitoring and station relocation in some environments, have resulted to irregular distribution

of data over time (gaps in time series) (Bodjrènou et al., 2023b). Some authors reported few weaknesses related to the quality of data collected. Danso et al. (2019) indicated that high winds and dust decrease the quality of recording sensors by carrying solid particles (grains of sand, plant debris) and settling down on the sensors. Other weaknesses include the irregular maintenance of sensors, lack of financial resources, lack of specialized technicians and

inaccessibility, and environmental risks of the sites (Bodjrènou et al., 2023b). For instance, in Benin, the installation and monitoring of some of measuring stations failed because of frequent flooding, bush fires, and low mobilization of significant investment (Amou *et al.,* 2022). Therefore, challenge looks to the differentiation of climate evolution and the evolution of factors resulting from human activities, as well as their individual and combined impacts on

water resources in these omitted environments.

This challenge has prompted researchers to evaluate different data from climate models, reanalysis and satellite data in order to identify qualified data for hydrological studies across these environments (Dembélé *et al.,* 2020). An effective solution will help avoid to refer to supplementary activities such as data complement methods which are generally applied to fill

in missing data, and interpolation methods used to have a regular data field in the basins. Among the data evaluated, reanalysis has been more screened recently in Benin. Reanalyses are defined as retrospective analysis data and their importance focused on their products that are based on the assimilation of a large number of in-situ and remotely sensed observations into an atmospheric general circulation model (AGCM) (Bodjrènou *et al*., 2023b; Reichle *et al.,* 2017).

Most studies focused on the verification of the estimation qualities of reanalyses on precipitation and temperature, and a few rare times radiation and wind, while very few studies are interested in the hydrological cycle mainly runoff, evapotranspiration (ETR), water table





depth (WTD). For instance, the studies conducted by Reichle *et al.* (2017) failed to consider whether other products are better than Modern-Era Retrospective Analysis for Research and

Applications (MERRA) products. In addition, the use of MERRA products may be limited to the small basins observed in Benin such as Mono-Couffo basin due to the coarse resolution of this reanalysis (0.5°x 0.625°). However; European Centre for Medium-range Weather Forecasts (ECMWF) produced a climate reanalysis ERA5, mainly its single-level version with 0.25° x 0.25° resolution (ERA5) and the land surface version with 0.1° x 0.1° resolution (ERA5-Land)

(Hersbach *et al.,* 2020; Muñoz-Sabater 2019). These reanalysis products combine observational and model data to produce an accurate description of past climate and hydrology.

The water cycle is made up of several elements including Runoff, Evapotranspiration (ETR), water table depth (WTD), and Soil water content (SWC). Runoff is one the most closely examined element in hydrological simulations. It is responsible for flooding downstream of the

basin and is highly controlled by the hydro-morphological characteristics of the basin soil (slope, roughness, permeability). ETR is a key component of the water balance. It represents the amount of rainwater that returns to the atmosphere in the form of gas and accounts for over 50% of the water entering the basin in the form of rain. WTD indicates the drying-up of water stored in the basin, and by deduction the severe impact of drought on water resources. SWC is

an essential variable in the Global Climate Observing System (GCOS) because of its impact on climate and hydrological processes through atmospheric feedbacks (Mahfouf, 2010). SWC is also a very important component for agronomists, enabling them to design an efficient agricultural calendar to optimize yields.

Several studies evaluating reanalysis in hydrology focused on meteorological variables

highlighted the potential of ERA5 reanalysis. Some reported that ERA5 is the best reanalysis product by comparing ERA5 meteorological data with meteorological data from observations (Grenier *et al.*, 2020, Danso *et al.,* 2019). Others, focusing on the outputs of hydrological simulations carried out with the ERA5 meteorological data, concluded that ERA5 enables better hydrological simulations. Bodjrènou *et al*. (2013c) reported better simulations with ERA5 in

the Ouémé basin (Benin Republic). In the north-western part of the same basin, mainly the Volta basin, Dembélé *et al.* (2020) presented the performance of ERA5 meteorological data on simulations of water cycle elements compared with observational data. None of these studies mentioned the performance of ERA5 reanalysis on these water cycle elements. Here, the current study aimed to fill this gap by considering the ERA5 reanalysis identified as the best and

performant, to validate or fine-tune hydrological models for the past period. This will be useful, especially for distributed hydrological models such as ParFlow-CLM, which can come out with



high-resolution simulations (spatially and temporally), but for which it is difficult to assess their quality on past simulations at each pixel.

In the remainder of this document, we presented the methodology, the results on the performance of runoff, evapotranspiration (ETR), water table depth (WTD) and soil water content (SWC) on these reanalysis, before concluding with a discussion.

## 2. Methodology

### 2.1. Study Environment

The present study was carried out in Benin, a West African country located on the Gulf of Guinea between parallels 6°30' and 12°30' and meridians 1° and 3°40'. It is dominated by a humid climate with mean annual rainfall of around 1.200 mm in the south and 1.100 mm in the north (Bodjrènou *et al.*, 2023b). Benin lies on the Atlantic Ocean (south) and is bordered by Togo (west), Burkina Faso (northwest), Niger (northeast), and Nigeria (east), with a surface area of around 114.763 km$^2$. It has four major watersheds: the Mono-Couffo (5.50% of the national surface area), the Volta (13.14%), the Niger (38.70%), and the Ouémé (41%) (Figure 1).

The Mono-Couffo basin is dominated by clay soils, mainly vertisol. The Ouémé basin is dominated by nixisol and Nitisol in the northeastern basement zone and southern sedimentary zone respectively (Lawin *et al.*, 2019). The Volta basin is dominated by a low-nutrient and unstable soil (Luvisol) that is conducive to erosion and the phenomenon of battance on sloping land (Mul *et al.*, 2015). The Niger basin is essentially bedrock, highly impermeable at depth, with granite and gneiss in the surface layer. It is home to reservoir construction projects (71% basement and 29% sediment) for agropastoral, domestic, swimming and other purposes (Sambieni *et al.*, 2023).



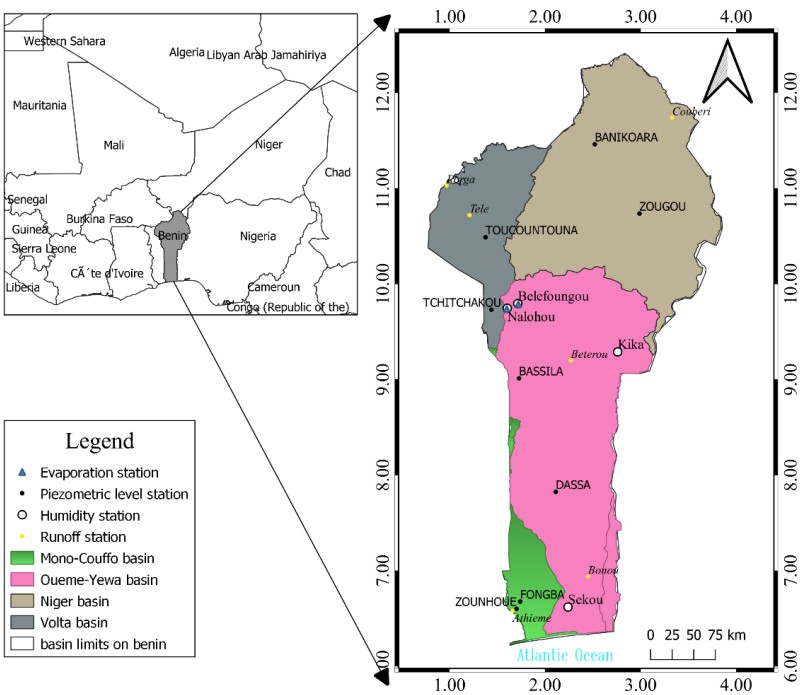

Figure 1: Map of the study area showing the Benin basins at the political boundaries

*Unlike soil water content stations (white dot) and evapotranspiration stations (blue triangle),*
*piezometric station (black dot) names are capitalized, while runoff station names (yellow dot)*
*are in italic.*

## 2.2. Data Collection

The hydrological data (runoff, ETR, WTD and SWC) were used in this study and came from
two categories including field-measured data and reanalysis data. Runoff data were collected
from the hydrometric stations that are best monitored and archived at the General Management
of Water (DGE). For each basin in the country, one station has been selected at least, covering
the period of 1980 to 2020 (Table 1). The WTD data were obtained from the DGE for the period
of 2000 to 2019. SWC and ETR data were povided from the AMMA-CATCH project (Analyse
Multidisciplinaire de la Mousson Africaine - Couplage de l'Atmosphère Tropicale et du Cycle
Hydrologique). These data have a very high temporal resolution (hours). Finally, the PAREA
project "Projet d'Appui à la Résilience des Entreprises Agricoles" provided us with SWC data
for the period of 2019 to 2020 (Bodjrènou *et al.*, 2022).

For the reanalysis data, we used two ERA5 products mainly the single-level version with 0.25°
x 0.25° resolution (ERA5) and the land surface version with 0.1° x 0.1° resolution (LAND)





which are freely available. Stations and the distances of the points selected in the reanalysis,
calculated from their longitude and latitude coordinates, are listed in Table 1.

Table 1: Data Used
*D1 and D2 indicate, in km respectively, the distance of the ERA5 and LAND pixel from the
point station coordinates. ETR indicates Evapotranspiration, WTD the Water Table Depth and*
*SWC the Soil Water Content.*

| | N° | Name | Lat. | Lon. | Period | D1 | D2 | Source | Ref. |
|---|---|---|---|---|---|---|---|---|---|
| **Runoff** | 1 | Coubéri | 3.33 | 11.74 | 1960-2017 | 8 | 5 | DGEAU | https://eau-mines.gouv.bj/structure/9/direction-generale-eau/ |
| | 2 | Porga | 0.97 | 11.03 | 1952-2013 | 4 | 4 | | |
| | 3 | Tele | 1.21 | 10.72 | 1961-2012 | 5 | 2 | | |
| | 4 | Bétérou | 2.27 | 9.2 | 1960-2020 | 5 | 3 | | |
| | 5 | Bonou | 2.45 | 6.94 | 1960-2019 | 8 | 7 | | |
| | 6 | Athiémé | 1.66 | 6.58 | 1980-2020 | 13 | 4 | | |
| **ETR** | 1 | Nalohou | 1.6 | 9.74 | 2007-2021 | 11 | 4 | AMMA | https://doi.org/10.17178/AMMA-CATCH.AE.H2OFlux_Odc |
| | 2 | Béléfoungou | 1.72 | 9.79 | 2007-2021 | 5 | 2 | | |
| **WTD** | 1 | Banikoara | 2.52, | 11.46, | 2009-2022 | 4 | 4 | DGEAU | https://eau-mines.gouv.bj/structure/9/direction-generale-eau/ |
| | 2 | Zougou | 2.99, | 10.74, | 2010-2020 | 1 | 4 | | |
| | 3 | Bassila | 1.73, | 9.01, | 2010-2022 | 2 | 3 | | |
| | 4 | Dassa | 2.11, | 7.83, | 2009-2022 | 15 | 3 | | |
| | 5 | Zounhoue | 1.70, | 6.60, | 2009-2023 | 12 | 0 | | |
| | 6 | Fongba | 1.74, | 6.68, | 2009-2023 | 8 | 5 | | |
| | 7 | Toucountouna | 1.38, | 10.49, | 2010-2022 | 13 | 2 | | |
| | 8 | Tchitchakou | 1.44, | 9.73, | 2009-2022 | 7 | 5 | | |
| **SWC** | 1 | Sekou | 2.24 | 6.62 | 2018-2020 | 13 | 4 | PAREA | Bodjrènou *et al* (2022) & https://doi.org/10.17178/AMMA-CATCH.CE.SW_Odc |
| | 2 | Kika | 2.76 | 9.29 | 2018-2020 | 4 | 4 | | |
| | 3 | Nalohou | 1.6 | 9.75 | 2007-2021 | 11 | 5 | AMMA | |
| | 4 | Béléfoungou | 1.71 | 9.8 | 2007-2021 | 6 | 1 | | |

## 2.3 Data Processing And Analysis

Point station data were first processed to eliminate biases and comply with the following
protocol: SWC values must lie between 0 and 1; runoff and water table depth data must be equal
or greater than zero; ETR values must be positive and not exceed 100% of rainfall.
On the reanalyses, only SWC data on the first three soil layers were used (layer 1: 0 - 7 cm,
layer 2: 7 - 28 cm, layer 3: 28 - 100 cm). Data were measured at depths of 10 cm, 20 cm and 60
cm and were compared with layer 1, layer 2 and layer 3 respectively. The pixels closest to the
observation data were chosen for comparison, both for SWC and for the other terms of the water
balance.





Metrics used for assessing the quality of reanalyses include Pearson correlation (c), Mean Absolute Error (m) and Relative Mean Absolute Error (r) as used by Bodjrènou *et al*. (2023b) and Gleixner *et al.* (2020). The formulas used for calculating c, m and r are respectively presented in Eq. 1, Eq. 2 and Eq. 3.


$$c = \frac{\sum_{i=1}^{N}(x_i - \bar{x}).(y_i - \bar{y})}{\sqrt{\sum_{i=1}^{N}(x_i - \bar{x})^2} . \sqrt{.\sum_{i=1}^{N}(y_i - \bar{y})^2}} \qquad \text{(Eq. 1)}$$

$$\text{m} = \frac{1}{n}\sum_{i=0}^{n}|Y_{rea} - Y_{obs}| \qquad \text{(Eq. 2)}$$

$$r = \frac{\sum_{i=0}^{n}|Y_{rea} - Y_{obs}|}{\sum_{i=0}^{n} Y_{obs}} = \frac{n * m}{\sum_{i=0}^{n} Y_{obs}} \qquad \text{Eq. 3}$$

With $Y_{obs}$ : the observation data, $Y_{rea}$: the reanalysis data and n the sample size

## 3. Results

### 3.1. Runoff Performance

Figure 2 shows the spatial variability of average annual runoff on both reanalyses over Benin and its performance at the six hydrometric stations included in this study. The average annual runoff varies between 0 and 0.2 m/h in the ERA5 compared with 0 and 0.9 m/h in ERA5-Land. The lowest values were generally reported above latitude 11° for both reanalyses. At this height,

ERA5 showed low runoff (approx. 0.08 m/h) and LAND showed no runoff (less than 0.05 m/h). LAND was therefore weaker than ERA5 in this environment, whereas it showed higher runoff below latitude 11°. The highest runoff values for LAND were reported in the sub-equatorial zone (latitude below 7°), probably due to the high precipitation that this reanalysis presents in this environment (Bodjrènou *et al.,* 2023). In addition, a high runoff (above 0.6 m/h) was

sometimes reported in the central-western part of the country, probably due to the influence of the Atacora mountain range.



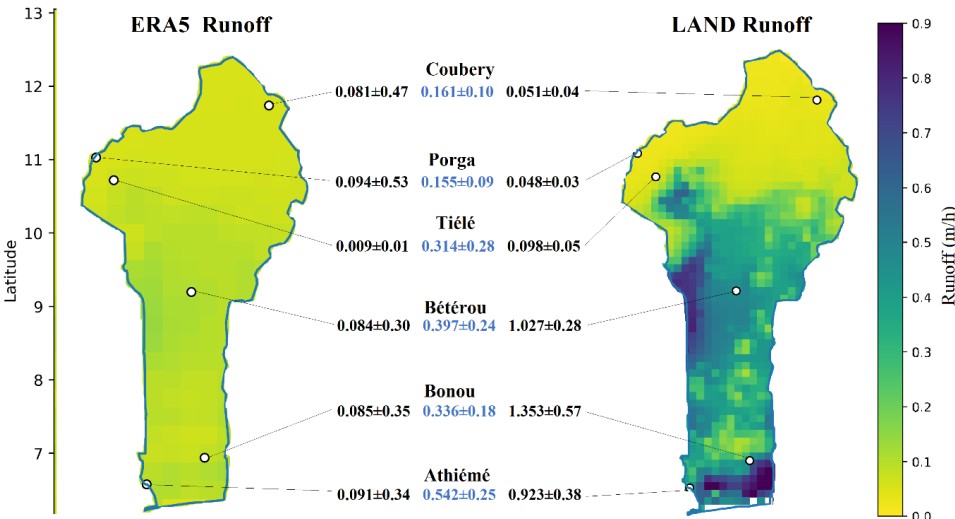

**Figure 2:** Spatial variability of runoff in Benin between 1980-2019 on ERA5 (left) and LAND (right). Mean annual values are shown directly below station names (observations in the middle in blue, ERA5 on the left and LAND on the right).

Results showed a decreasing south-north gradient between 0.542±0.25 m/h (Athiémé station) and 0.155±0.09 m/h (Porga station). The reanalyses showed very large differences in average annual runoff in the south (m=0.54 for ERA5 and m=0.49 for LAND at Athiémé) compared with the north (m=0.24 for ERA5 and m=0.11 for LAND at Coubéri). Pearson correlation coefficients remained better for LAND than for both ERA5 and m (Figure 3).

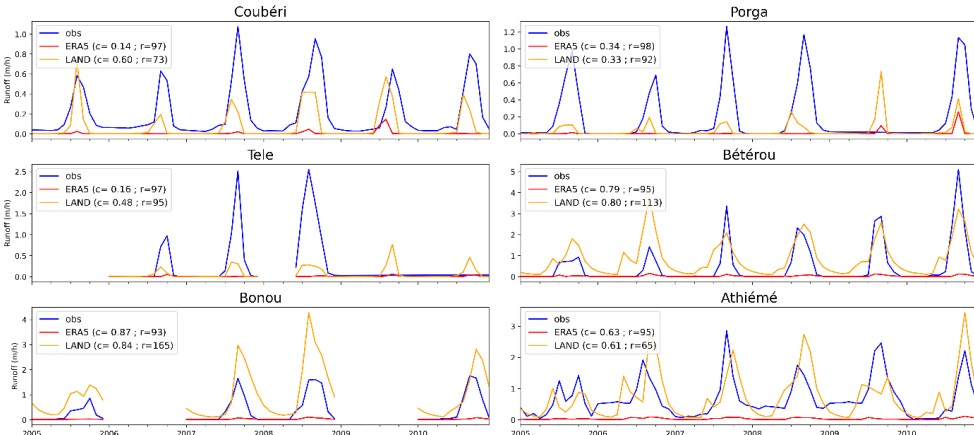

**Figure 3:** Reanalysis performance on mean monthly runoff variability between 2005-2010. Letter $c$ indicates Pearson correlation and letter r the Relative Mean Absolute Error value between observations (in blue) and reanalyses (in red for ERA5 and orange for LAND).





Maximum annual runoff (not shown) presented the better correlation on LAND (0.61) compared to ERA5 (0.33).

Figure 3 shows the variability of mean monthly runoff at the six hydrometric stations previously
indicated. Analysis of this figure reveals the average monthly runoff peak in August-September-October (ASO). During the months of January-February-March (JFM), observation data showed zero runoff, except at certain stations such as Athiémé.

The runoff peaks on the reanalyses are similar to the observation peaks. There is, however, a one-month difference between peaks. This is the case for the Coubéri station, where both
reanalyses showed their peak for 2009 in August, whereas the peak in the observations was reported in September. Both reanalyses underestimated average monthly runoff at all hydrometric stations, especially during peak periods (ASO). However, underestimates mean monthly runoff in certain years at stations such as Bonou were reported for LAND during the same period. Correlation obtained with LAND was about 0.61 with lowest biases at most
stations (four out of six: Coubély, Tiélé, Porga and Athiémé), compared with ERA5 where correlation obtained was about 0.49.

### 3.2. Water Table Depth (WTD) Performance

Figure 4 shows the spatial variability of WTD in Benin over the two reanalyses, with the
precision of annual mean values at eight (08) well-monitored piezometric stations. Both reanalyses showed WTD equal to zero (00)m at certain points, including the Niger-Benin border in the northeast of the country (probably due to the presence of the Niger River, which is a perennial river), and in the south for LAND, probably due to the presence of Lake Nokoué.





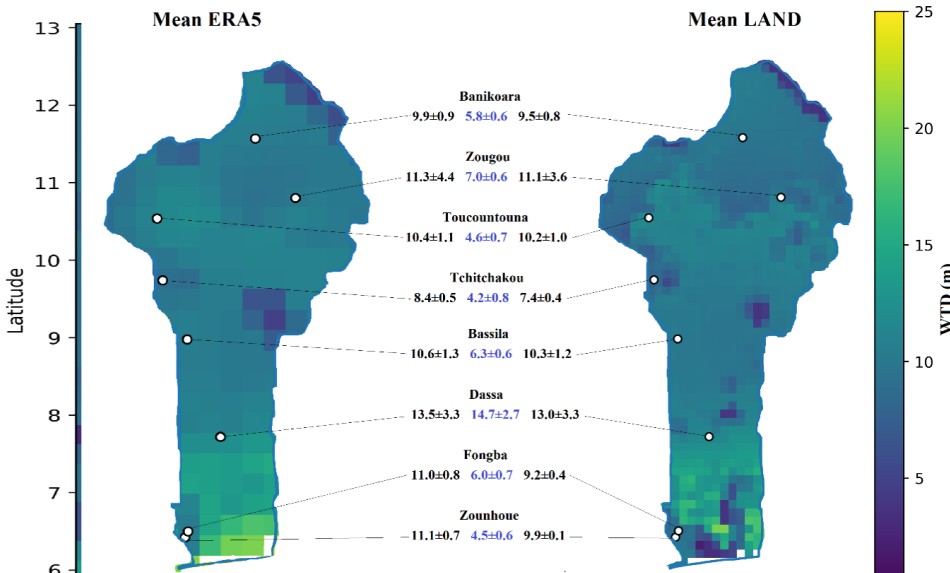

**Figure 4:** Spatial variability of water table depth (WTD) in Benin between 1980-2019 at ERA5 (left) and LAND (right). Mean annual values are shown directly below station names (observations in the middle in blue, ERA5 on the left and LAND on the right).

Point station data (in blue) at eight stations indicate WTD ranging from 4.4m (Tchitchakou station) to 14.7m (Dassa-Zoumè station). Groundwater appears to be more accessible in the south than in the center and north. Reanalyses showed average annual values more or less consistent with observations. For instance, observation data from the Tchitchakou station showed the shallowest WTD of the eight (08) stations (4.4±0.8m). The values on reanalyses (8.4±0.5m for ERA5 and 7.4±0.4m for LAND) consistent with observation were aligned with this assertion. The same trends were reported for the Dassa-Zoumè station, which has the deepest WTD of the eight (08) stations, both in terms of observations (14.7±2.7m) and reanalyses (13.5±3.3m for ERA5 and 13.0±3.3m for LAND). With the exception of this station, it is observed that the reanalyses overestimate the overall WTD at the study stations. While the observations showed averages of 6.6±0.9m, we reported average WTD of 10.8±1.6m at ERA5 and 10.1±1.4m at LAND, probably due to errors in the Digital Terrain Models.

Figure 5 shows the variability of the monthly cycle of the WTD at the eight stations. The observations (in blue) show that the monthly cycle of the WTD is almost sinusoidal at the Banikoara, Dassa-Zoumè, Tchitchakou and Toukountouna stations, in contrast to the other stations, which have almost linear cycles, probably linked to the soil profiles. At the Banikoara

station (Founougo), the WTD is lowered from November/December to July/August. September and October are months of recharge. The situation is almost identical at all stations.

Both reanalyses showed a sinusoidal cycle at all stations. With the exception of the Dassa-Zoumè (Mahu) station, where they showed a bimodal cycle (a first drawdown peak between December-January: period of the long dry season in the south, and a second drawdown peak between August-September: period of the short dry season in the south), the other stations show a single drawdown peak between December-January. These cycles revealed by reanalysis often not in phase with those of observations, leading to low correlation values. The reanalyses simulate average WTD that are approximately equal to those observed. Slight overestimates are 4.73m on ERA5 and 3.13m on LAND. Large differences were reported at the Zounhouè station (6.94m for ERA5 and 5.93m for LAND), probably due to its proximity to the Mono River.

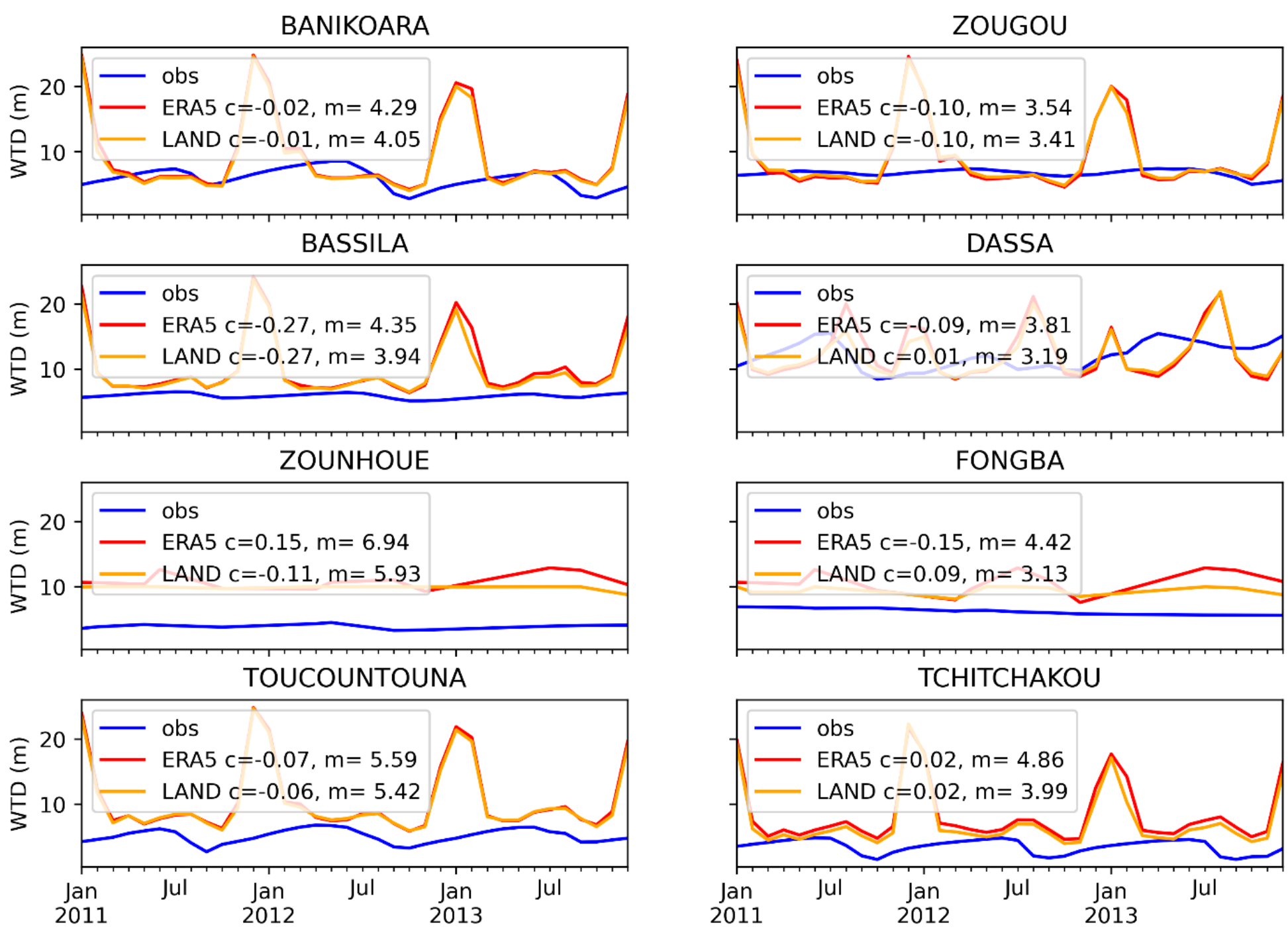

Figure 5: Reanalysis performance on mean monthly WTD variability between 2005-2010. c indicates the Pearson correlation and m the Mean Absolute Error value between observations (in blue) and reanalyses (in red for ERA5 and orange for LAND).





### 3.3. Soil Water Content (SWC) Performance

Figure 6 shows the spatial variability of mean SWC between 0 -7 cm soil depth in Benin, based on the ERA5 and LAND reanalyses. Results reveal that the two reanalyses showed soils that are generally less moist in the extreme north of the country (latitude above 11°), with an average

of 17.90% and 18.23% respectively for ERA5 and LAND, compared with the south (latitude below 11°: average ERA5=30.46% and LAND=30.23%). Same trends were reported for the deeper soil layers (see appendix, fig. A1). Point station data measured at 10 cm depth at the Béléfoungou and Nalohou stations showed values of 11.1±5.8% and 11.6±7.4% respectively. At a depth of 60 cm, mean values of 24.5±8.1% and 31.8±9.7% were recorded for the same

stations (Béléfoungou and Nalohou respectively), indicating an increase in SWC with depth. ERA5 and LAND also showed increases with depth at both stations, with the exception of Béléfoungou for LAND. Both reanalyses overestimate SWC at almost all stations, for both the first layer (0-7 cm) and the second layer (7-28 cm).

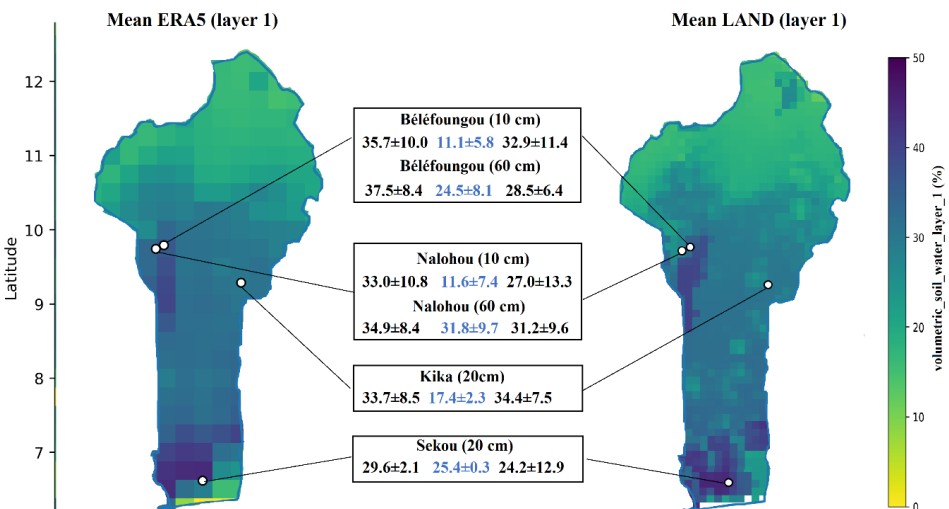

Figure 6: Spatial variability of soil water content in Benin between 2000-2010 at ERA5 (left) and LAND (right). Mean annual values are shown directly below station names (observations in the middle in blue, ERA5 on the left and LAND on the right).

Figure 7 shows the performance of reanalyses on monthly soil water content (SWC) variability
at selected stations in Benin. At the Béléfoungou and Nalohou stations, the observation data (in blue) showed that SWC is almost nil during the long dry season, mainly in the months of December-January. The monthly cycle is almost sinusoidal at both stations, with a peak in August-September-October (ASO), which corresponds to the peak of the rainy season in this





area. Both reanalyses show cycles consistent with observations, leading to correlations greater
than 0.80. However, they considerably overestimate SWC at both stations, with higher values
at ERA5 (24% and 21%) than at LAND (21% and 15%) for Béléfoungou and Nalohou
respectively. The situation at Kika is almost identical to the last two, probably because this
station is also located in the Sudano-Guinean zone, where the monthly cycle is unimodal
(Bodjrènou *et al.,* 2023a). At Sekou station, LAND's monthly cycle is bimodal, more in line
with observations (correlation equal to 0.56) than with ERA5 (correlation equal to -0.52).

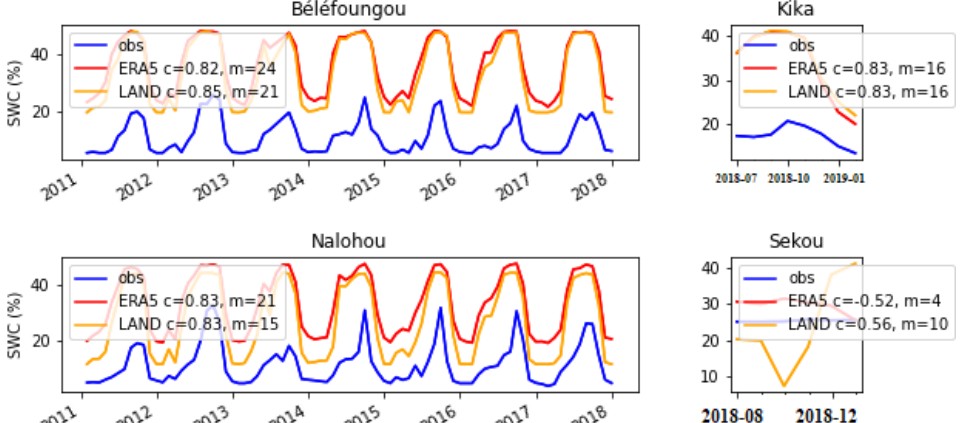

Figure 7: Reanalysis performance on mean monthly humidity variability between 2005-2010.
c indicates the Pearson correlation and m the Mean Absolute Error value between observations
(in blue) and reanalyses (in red for ERA5 and orange for LAND).


### 3.4. Evapotranspiration (ETR) Performance

Figure 8 shows the spatial variability of ETR in Benin for the ERA5 and LAND reanalyses.
Analysis of this figure reveals that evapotranspiration is lower in the north (latitude above 11°)
for both ERA5 (1.94 mm/day) and LAND (0.88 mm/day) than in the south (latitude below 11°),
where ERA5=2.51 mm/day and LAND=1.27 mm/day. The Béléfoungou (higher latitude) and
Nalohou (lower latitude) stations do not seem to respect this principle, probably due to the
influence of the Atacora chain (Bodjrènou *et al.,* 2023). In terms of observations, the reported
ETR is slightly higher at the Béléfoungou station (3.34±2.36 mm/day) than at the Nalohou
station (3.15±1.70 mm/day). Both reanalyses showed slight underestimations of ETR at both
stations, but following the same gradients in values, with Béléfoungou higher (2.94±0.88
mm/day for ERA5 and 1.50±0.51 mm/day for LAND) and Nalohou lower (ERA5=2.90±0.91
vs LAND=1.49±0.46 mm/day).



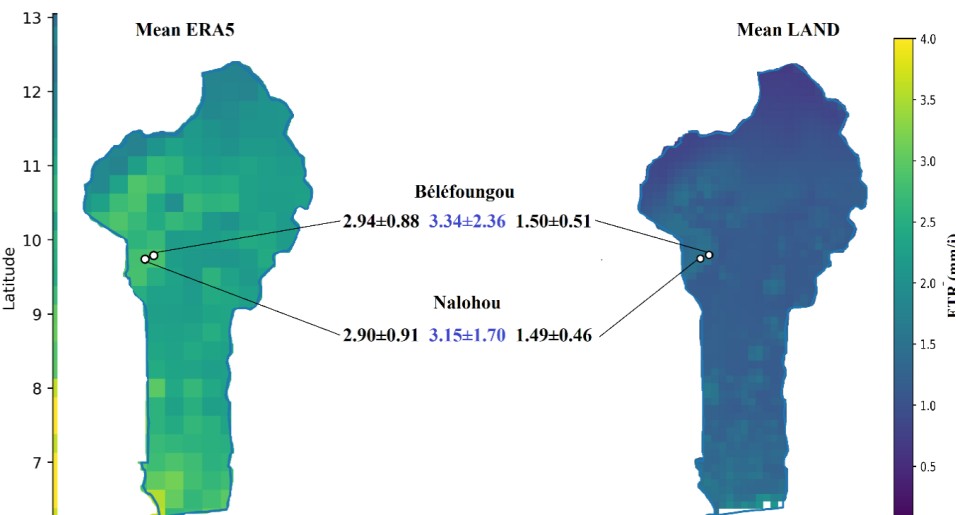

Figure 8: Spatial variability of actual evapotranspiration in Benin between 1980-2019 on ERA5
(left) and LAND (right). Annual mean values are shown directly below station names
(observations in the middle in blue, ERA5 on the left and LAND on the right).

Figure 9 shows the monthly cycle of evapotranspiration (ETR) at the Nalohou and Béléfoungou
stations. From the observations (in blue), ETR is generally low between December and January
at both stations. Overall, August, September and October (ASO) are the peak months for ETR
at both stations.

Both reanalyses showed monthly averages consistent with observation data. This is evidenced
by correlation values consistently above 0.5 for both ERA5 and LAND. Furthermore, both
reanalyses underestimated ETR in ASO comparatively to December and January. However,
mean ETR was better simulated on ERA5 at Nalohou (m =1.05, corresponding to r=33%) than
on LAND at the same station (m=1.81mm/day, corresponding to r=57%). Performance was
almost identical at the Béléfoungou station.




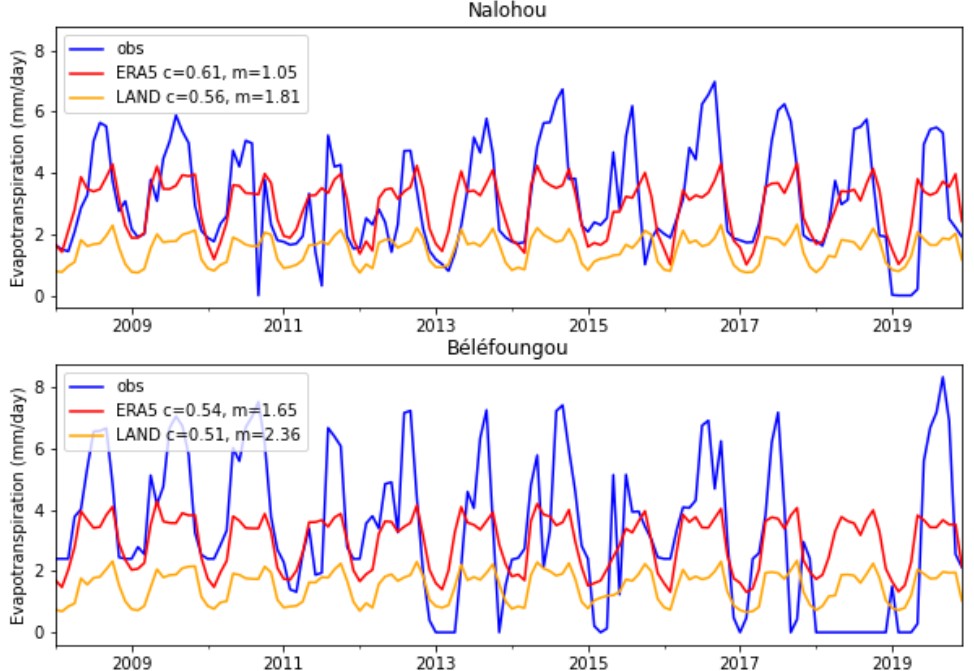

Figure 9: Performance of reanalyses on variability of monthly mean evapotranspiration between 2005-2010. c indicates the Pearson correlation and m the Mean Absolute Error value between observations (in blue) and reanalyses (in red for ERA5 and orange for LAND).


## 3.5 Checking the Hydrological Cycle Closing

The Table 2 shows that neither ERA5 nor LAND has a closed balance. For example, results on ERA5 indicated that in 2014, ETR represented 77% of precipitation and therefore storage is 0%, and the runoff represents 25%. The total output has been estimated at 102% of precipitation (only input). On LAND, the ETR represents 39% and runoff 9%, this corresponds to 48% as total output. We can therefore recommend that the reanalyses be corrected by variable and by fault.


Table 2: Hydrological balance between 2010 and 2019 on ERA5 and LAND.
*The values given in the table for each of the water balance terms are in mm/year.*

| Reanalyses | Term | 2010 | 2011 | 2012 | 2013 | 2014 | 2015 | 2016 | 2017 | 2018 | 2019 |
|---|---|---|---|---|---|---|---|---|---|---|---|
| | Rain | 1109 | 899 | 1281 | 965 | 1158 | 1016 | 1217 | 1040 | 1170 | 1270 |
| | ETR | 864 | 821 | 882 | 882 | 897 | 834 | 889 | 857 | 870 | 897 |
| ERA5 | Runoff | 280 | 184 | 346 | 192 | 293 | 215 | 348 | 266 | 325 | 394 |
| | Storage | 28 | 27 | 29 | 28 | 28 | 27 | 28 | 27 | 28 | 29 |





| | | | | | | | | | | |
|---|---|---|---|---|---|---|---|---|---|---|
| | Rain | 1112 | 894 | 1289 | 964 | 1154 | 1015 | 1222 | 1040 | 1172 | 1270 |
| | ETR | 427 | 401 | 448 | 435 | 447 | 423 | 447 | 419 | 432 | 447 |
| LAND | Runoff | 109 | 74 | 134 | 66 | 108 | 76 | 139 | 95 | 124 | 148 |
| | Storage | 27 | 26 | 28 | 27 | 27 | 27 | 28 | 27 | 27 | 28 |


## 4. Discussion And Conclusion

This study evaluated the hydrological estimation of the ERA5 reanalysis over Benin. Both versions of the reanalysis including ERA5 single levels version and ERA5 at the land surface

called LAND, were screened. The variables highlighted include runoff, evapotranspiration (ETR), soil water content (SWC) and water table depth (WTD).

### 4.1 Limitations of the Study Methodology

Spatially, the reanalysis data were evaluated on the basis of the observation data, which are collected on point station. If the values indicated in the reanalyses stand for the station

coordinates, it is possible that data compared in this study are a few kilometers apart. Results were based on the comparison of spatialized data that stand for data averaged over a given scale, and the observations used do not consider the same scales. In addition, the distances between the observation stations and the nearest pixels are not identical. This constitutes a real limitation on the evaluation method of variables, especially runoff and water content. With regard to the

limitations of the comparison of ETP stations and, to a lesser extent, water table depth stations (WTP), readers may refer to the study of Bodjrènou *et al.* (2023b).

### 4.2. Discussion on Runoff

Runoff is defined in reanalyses as the portion of water originating from precipitation, snowmelt or deep ground that runoff either over the surface (surface runoff) or under the ground

(subsurface runoff). They express it as the height the water could have reached if it were uniformly distributed over the grid. Analysis of the spatial variability of runoff over the whole country gives the impression that there are no rivers in Benin. This may be due to the resolution of the reanalyses discussed above. Bodjrènou *et al* (2023b) explained that Benin's hydrometric stations are installed in waterbeds with ceilings measuring a few meters (all less than 1 km).

However, here, data were compared with reanalysis data, which are spatialized over more than 10 km for LAND and more than 25 km for ERA5. This could explain the underestimates





reported in the reanalyses and the strong underestimates on ERA5, associated with its coarser spatial resolution. Furthermore, the observations are in m³/hour, which has been broken down by the surface area of each basin in order to bring them back to the same unit with the

reanalyses. The surface area considered in this operation is nothing other than the surface area of the hydrological basin and not the surface area of the hydrogeological basin, which is generally larger than the former. Reanalyses sometimes show negative values, depending on whether the water is runoff in the opposite direction to the one considered and/or is rising by capillary action. These elements are not considered in the observation measurements resulting

in positive mean values.

On the monthly cycle, our results showed that runoff peaks in September, probably because this is the heart of the rainy season in the north, and runoff are generally one-way (north to south). Results also showed that runoff is often nil between December and January at hydrometric stations, apart from a few such as Athiémé where runoff are non-zero during this period. These

results are in line with studies by Amou et al. (2022) which indicated the absence of dry-season runoff in the Oueme basin at Bonou. For Giertz *et al.* (2010), hortonien runoff is sometimes observed in this basin, indicating the complicity of the basin's hydrological system, making it difficult for these measurements to be accurate. For the ERA5 reanalysis, results showed a strong underestimation, probably due to its coarser spatial resolution. Considering that the near-

surface water height of the ERA5 reanalysis was reported at 1° resolution, a considerable improvement in the performance of this reanalysis was reported. This proves that this reanalysis is not necessarily the least efficient, but there are significant biases due to its resolution, or even the way in which the water height is estimated in the reanalyses. In-depth analyses have shown that the distribution in this reanalysis presents very extreme runoff values on a daily scale

(maximum ERA5=23.3m/h vs. 8.3m/h for observation and 7.5m/h for LAND).

The highest runoff values for LAND reanalysis were reported in the sub-equatorial zone (latitude below 7°), probably due to the high precipitation that this reanalysis presents in this environment (Bodjrènou *et al.,* 2023). Similarly, high runoff (above 0.6 m/h) is sometimes observed in the central-western part of the country, probably due to the influence of the Atacora

mountain range. LAND is the best-performing reanalysis in terms of correlation (~0.61) compared with ERA5 (correlation ~0.49).



### 4.3. Discussion on Water Table Depth (WTD)

Reanalyses define WTD as the average depth of groundwater bodies (lakes, reservoirs, rivers and coastal waters) across the globe, even where there is no inland water. The spatial variability maps presented in our results show zero values in the large permanent water bodies environments of Benin, which is consistent with the definition. Although ERA5 did not show zero values in all the places where LAND identified water at the surface, showing low water

table levels in the majority of points, due to the extent of the water bodies. This is the case for the River Niger, which is 250 km wide on average, covering almost 25 pixels of LAND but around 10 pixels of ERA5.

On the monthly cycle, there were significant differences in the WTD at Zounhouè station (6.94 m for ERA5 and 5.93 m for LAND), probably due to its proximity with Mono River, which

runoff in a bed only a few meters wide at Athiémé, which escapes the digital field model of the two reanalyses. There was little correlation between the monthly cycle of the reanalyses and that of the observations, probably because the reanalyses have no means of considering the variation in aquifer water levels, other than the upper limit of the thermocline located at the bottom of the mixed layer and the lower limit of the thermocline located at the bottom of the

lake. Thus, the variation in WTD can be influenced by precipitation in reanalyses, whereas it takes time for rainwater reaching the surface to reach the water table. High average WTD values were reported for reanalyses compared with observations. This may also be due to errors in the Digital Field Models. This study did not analyze the variability of water levels at permanent points. It would have been interesting to do so in order to quantify the volume of water available

at the surface throughout the country, but we have no data for comparison.

### 4.3. Discussion on Soil Water Content (SWC)

SWC is defined in the reanalyses as the volume of water in the soil layer. It is masked on regions with a water surface, considering that grid points where the land-sea mask has a value greater than 50%. This justifies the fact that areas of permanent water presence are not saturated

(normally 100% value). In addition, SWC in reanalyses is associated with soil texture. Today, there are standard soil classification norms. This constraint could force us to choose the most dominant class among those identified in the reanalyses, thus accepting to neglect certain important formations in the basins (Amoussou, 2010). This certainly justifies the large differences in mean values reported between observations and reanalyses. In addition, this may

be also due to the spatial average of the water content considered in the models compared with





the point data as already mentioned above. The smallest biases were observed at the Sékou station for both reanalyses, probably because the station is located within a vegetable garden of the secondary school of Agriculture, where water inputs are more or less homogeneous, as are organic matter inputs, and vegetation is almost evenly distributed (Bodjrènou *et al.,* 2022).

Results indicated that the spatial variability of SWC in the first layer (0-7) in reanalyses above latitude 11° was lower than below this latitude. SWC in the first layer near the surface was strongly influenced by temperature variability. Thus, the decrease in SWC at this height can be explained by the high temperature variability above this altitude (Bodjrènou *et al*. (2023b)).

With regard to the variability of the monthly cycle, a good phasing between the evolution of
the reanalysis and the observations was reported, because the reanalysis describes well the monthly temperature cycle, which really influences SWC (Bodjrènou *et al.*, 2023b). It was also found that the SWC measured at 60 cm on the Nalohou station was moderately closer to the SWC in the second layer of the two reanalyses. At the same time, the measurement at 40 cm, which was supposed to be closer to the average in this layer, was more biased. These results
may be associated firstly with the variation in the digital field model, and with the fact that the model considers an average for each layer of soil thickness on the other hand (Amou *et al.,* 2022).

### 4.4. Discussion on Evapotranspiration (ETR)

Reanalyses define this parameter as the accumulated amount of water that has evaporated from
the land surface, including a simplified representation of transpiration (from vegetation). Our observation stations did not distinguish between evapotranspiration (ETR) from the surface and ETR over vegetation, but they are well instrumented to present the accumulation of the two that represents ETR as defined in the reanalyses.

Results showed that spatial variability on reanalyses is lower above 11° latitude than below 11°
latitude. This can be explained by the higher temperature (Bodjrènou *et al.,* 2023b), added to the less dense vegetation above this height (Bodjrènou et al., 2023c). In the same vein, Awessou *et al.* (2017) in a comparative study between the transpiration of a forest species and an agroforestry species, reported that evapotranspiration in forests is linked not only to the density of shrubs, but also to their composition. This probably justifies the higher ETR observed at
Béléfoungou (forest zone) than at Nalohou (fallow zone). The higher standard deviation value for the latter compared with the Nalohou station may be explained by the fall of foliage in the dry season and its recovery in the rainy season. The standard deviation gradients also seem to

be respected in LAND, probably also due to its less coarse homogeneous resolution of the Nalohou site in terms of vegetation (Bodjrènou *et al.,* 2023c).

In terms of monthly cycle variability, ETR was generally low between December and peaks were reported at both stations for January and ASO. These results are in the same trends with the results of Mamadou *et al.* (2016) reporting less evapotranspiration in the dry season and high evapotranspiration in ASO. The low biases reported on ERA5 is a promising result encouraging the use of this reanalysis instead of LAND.

Result didn't highlight any best reanalyse, both spatially and on a temporal scale (monthly). To calibrate/validate the distributed models, further studies can rely on reanalyses based on their performance and the simulated water balance term. Correcting the variables of these reanalyses could improve their performance.

## Appendix

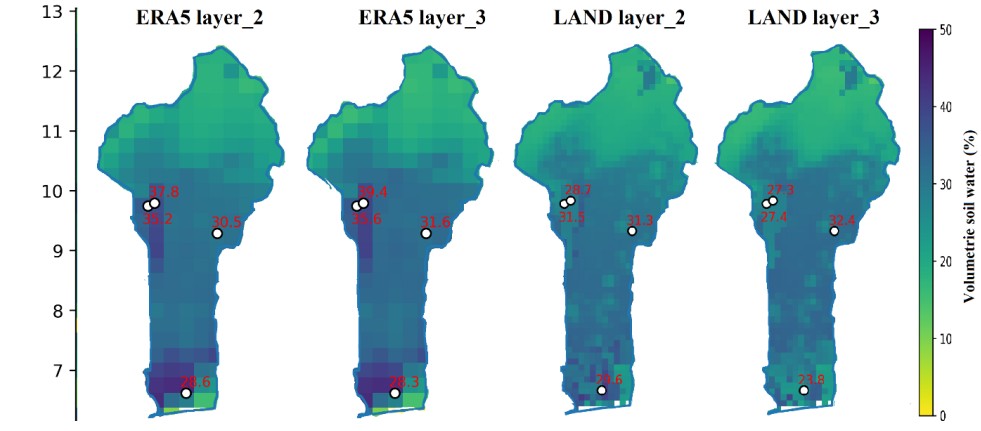


**Fig A1:** Volumetric soil water content in Benin on ERA5 and LAND on layers 2 and 3 between 2010-2019
*Values in red indicate the annual average at the measuring stations.*

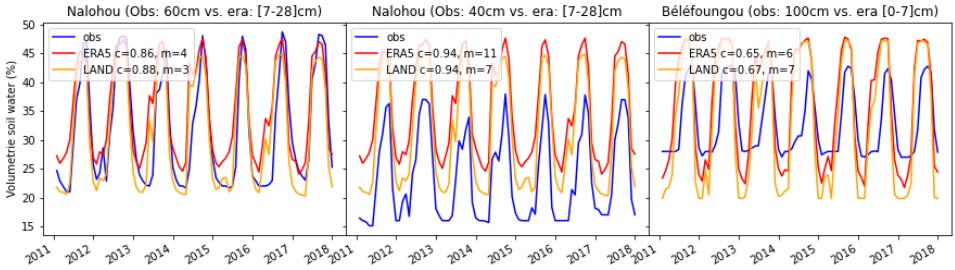


**Fig. A2:** Monthly changes in humidity at Nalohou (left: measured at 60 cm; middle: measured at 40 cm) and Béléfoungou (right). Observations in blue, ERA5 in red and LAND in orange.





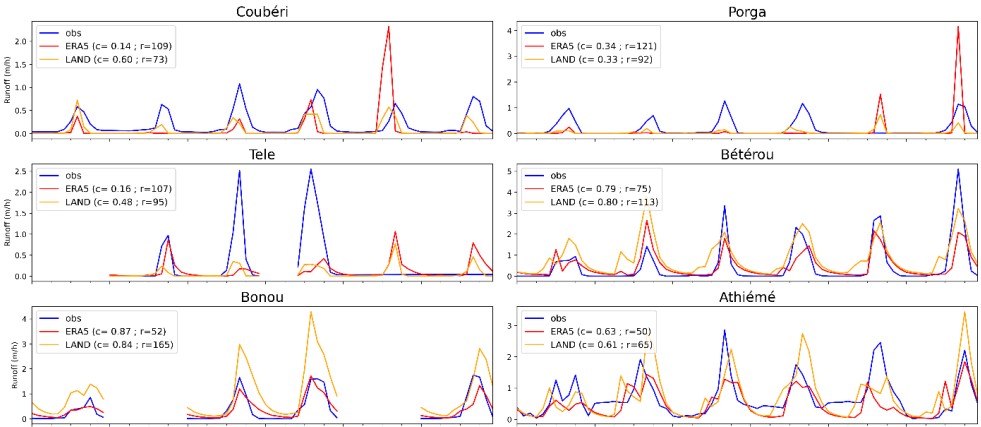

**Fig.A3:** Monthly runoff corrected on ERA5 assuming water is uniform over 1°.
Observations are in blue, ERA5 in red and LAND in orange. C=correlation and r =R

**5. Competing interests**

The contact author has declared that none of the authors has any competing interests

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
