# Peer review of "Assessment of Hydrology Estimates from ERA5 Reanalyses in Benin (West Africa)"

_Hydrology and Earth System Sciences, 2023_

## Author Comment (AC1)

**Assessment of Hydrology Estimates from ERA5 Reanalyses in Benin (West Africa)**

By : Bodjrènou et al. (reference: hess-2023-311)

**Responses to Anonymous Referee #1**

*We deeply thank the* anonymous *reviewer for his time and* constructive revisions, comments and questions *on the manuscript. In the revised version of the manuscript, we will consider all the comments and suggestions made by the reviewer. We strongly believe that the paper will be greatly improved and hope that the reviewer will agree with changes made.*

Before revising the article, we would like to briefly clarify the fundamental comments released by the reviewer

**RC1 - Comment 1)** The first concern is the novelty of the research. The primary approach of comparing ERA5 and ERA5-Land data at a regional scale with point station data is a method that has been extensively and comprehensively explored in the literature (e.g., https://doi.org/10.5194/hess-25-17-2021, https://doi.org/10.1016/j.jhydrol.2024.130649). The manuscript does not provide a sufficiently novel contribution to the body of knowledge that would justify its publication. The study follows a very straightforward assessment methodology without introducing new insights into the assessment process, analytical techniques, or applications that could significantly benefit the scientific community or advance the field of hydrological modeling in West Africa. For example, the main conclusion "This study does not identify any reanalysis that is better than another, both spatially and monthly scale. Nevertheless, this study indicated that the choice of reanalyses must rely on their performance and the given water cycle element. Correcting the variables of these reanalysis could also improve their performance." in the abstract is somehow well-known principle in hydrology science.

**Answer of authors:** Firstly, the reviewer states that the method used has been thoroughly and exhaustively explored in the literature and that this publication should introduce new knowledge about the assessment process, analysis techniques or applications that could significantly benefit the scientific community or advance the field of hydrological modelling in West Africa.

We believe that the novelty of an article cannot be judged solely by introducing a new method. It is possible to use an existing method given the data available and the objectives of the study. In the case of our study, the main aim is not to compare methods, nor to invent a method of comparison. However, we are open to any other proposal for evaluating and analyzing the data that the reviewer may put to us. We would also like to point out that, unlike the studies that deal with the evaluation of meteorological data from reanalyses, the evaluation of hydrological data, and in particular all the terms of the hydrological balance, is very rare and very little explored. Even less so, since this is the first time the study has been carried out in the study area.

With regard to the main conclusion cited by the reviewer as an example, we agree that this set of sentences can be a little confusing. We will make sure care to reformulate these sentences in the revised version. In advance, we would like to share with you the table below, which shows the reanalysis judged best according to the scale. In light of this result, we cannot describe a re-analysis as better if we are on a spatial scale. Similarly, we cannot conclude that ERA5 is better or worse at the temporal scale for all the variables. However, as indicated above, we will make sure to rephrase these sentences.

We might suggest using ERA5 to calibrate models on ETR and Runoff. But we suggest LAND for SWS and WTD, as indicated in the following summary: https://doi.org/10.5194/egusphere-egu24-12160

| Scale | Runoff | ETR | WTD | SWC |
|---|---|---|---|---|
| Spatial | ERA5 | ERA5 | LAND | LAND |
| Temporal | ERA5 | ERA5 | LAND | LAND |

**RC1 - Comment 2**) My second concern or curiosity is about the method of directly comparing ERA5(-LAND) runoff data, which does not include routing, with observed discharge data. The conclusion such as "Analysis of the spatial variability of runoff over the whole country gives the impression that there are no rivers in Benin" is weird because fundamentally, the runoff products are unrouted data that do not consider the lateral connectivity of grid cells through the river channel. To make a more meaningful comparison between the routed runoff and observed "discharge/streamflow," I recommend the authors refer to https://doi.org/10.5194/essd-12-2043-2020.

**Answer of authors**: We're flexible about using another method to assess throughput, but we think it's ideal to use the present method, which is based on the principle of data comparison. In this way, a unique method is used for the analysis of all variables. The second reason which justifies the choice of this method for the evaluation of the Runoff is as follows. We discovered in the data source that:

« The units of runoff are depth in meters of water. This is the depth the water would have if it were spread evenly over the grid box. Care should be taken when comparing model parameters with observations, because observations are often local to a particular point rather than averaged over a grid box. Observations are also often taken in different units, such as mm/day, rather than the accumulated meters produced here ».

This description led us to reduce the observation values to the same unit as the reanalysis data. In the case of our study, the data presented for each basin outlet is in $m^3$/day. They are converted to m/h by dividing by the surface area of the basin and the corresponding conversion factors.

The description of the reanalysis flow is available on the following link:

https://cds.climate.copernicus.eu/cdsapp#!/dataset/reanalysis-era5-single-levels?tab=overview

The reviewer also indicated that he had concerns about certain parts of the article. He referred to section 3.5. In his opinion, the annual verification of the balance is insufficient. First of all, we want to stress the importance of presenting the closure of the water balance. In fact, it is necessary in our study to decide on the choice of all the variables in any reanalysis. In the case of our study, we noted that neither the ERA5 nor the LAND balance is closed. Consequently, in future hydrological studies in this environment, we will be able to calibrate our model with the Runoff and ETR data from ERA5. However, the model can only be calibrated with LAND's water table depth (WTD) and Soil Water Content (SWC) data, as LAND appears to be the best performing reanalysis on both the spatial and temporal scales (see table above).

What's more, as the balance is not closed, there are no consequences in terms of improving the performance of the reanalyses on a specific variable by applying correction methods.

It should be pointed out that we have verified closure on an annual scale, and we feel that this is already essential if we are to draw the right conclusions. This is the scale on which many research projects focus

in order to verify the water stock. If not, we would have chosen to present the balance on a monthly scale, especially as our study evaluated the reanalyses on a monthly time scale. We are always flexible your decisions/recommendations.

The reviewer also wonders about the nature of the water stock in the soil. The word "stock" used here is a misuse of language. We have presented the soil water content and not the stock. This word will be corrected in the next submission.

Finally, the reviewer would like to know whether layer 4 is considered in the table or whether it has been excluded as in other parts of the article. Although we have excluded this layer in the other parts of the article, we have considered it here because there are no constraints. In the case of the other parts, it is because of insufficient data that we have excluded it.

---

## Author Comment (AC2)

**Assessment of Hydrology Estimates from ERA5 Reanalyses in Benin (West Africa)**

**Bodjrènou et al. (Reference: hess-2023-311)**

Before responding to Jean-Martial COHARD's concerns, I would first like to thank him sincerely for his precise, clear and constructive comments. I would also like to thank him for his interest in presenting very rich and useful research to the scientific community, and much more so to researchers in Benin. I trust that the scientific rigor he demonstrated in a similar article we published together (https://doi.org/10.1175/JAMC-D-21-0222.1) and his willingness to follow this article through to publication will help us to improve the quality of the manuscript.

1) a lack of references to back up numerous assertions throughout the text, use of unadapted citations, and a strong tendency to self-citation.

The author pointed out that the article contains statements without references and inappropriate quotes. It would have been better to have given two examples to guide us, but we take note of this comment and promise to check the article thoroughly.

Regarding self-citation, it should be  noted that two of our works must be cited in view of the interest they bring to this study. The first is to provide information about the soil moisture data that we collected as part of the "Projet d'Appui à la Résilience des Entreprises Agricoles (PAREA)". The second (published together with Cohard Jean-Martial) largely presented the limitations of comparing spatial data from reanalyses with point data. We have no problem referring to other studies that we have carried out in the study environment where the evaluation of reanalyses is very new, but we will take this comment into account.

2) Easy assertions, like this product "is the best analysis ..." without giving any criteria or metric to justify.

Criteria or metrics are provided for both spatial variability and the monthly cycle. Either the average values of the reanalyses compared with the observations are indicated, or the maximum peaks, correlation values or MAE are indicated. For example, we can mention the Runoff for which we indicated that MAE=0.54 for ERA5 and MAE=0.49 for LAND at Athiémé. However, we will check whether it is necessary to present metrics in certain other places in the document.

3) The methodology of the comparison is not introduced nor discussed. It seems that authors directly compared runoff at the river station (m/h) with the co-located pixel runoff variable from the ERA5 product, which is supposed to be given in mm, and which doesn't receive the runoff from the all drainage area because ERA5 is not routed. No information is given about the way this comparison has been processed.

Moreover, the runoff comparison concerns 4 main rivers in the Benin region which are all transboundary rivers. Incoming runoff from side countries is never evoked or discussed as authors just focus on the Benin territory. This is the same for the water table depth which is compared with the ERA5 product (I guess it concerns the sub-surface runoff variable expressed in m but it is not precised). To my knowledge, ERA5 doesn't provide any water table depth. The way the data has been processed to make this comparison possible is not detailed. Last, the authors seem to confuse PET with AET. The observations in figure 9 are clearly PET which are compared with AET from ERA5. By the way, the AMMA-CATCH observatory hasn't provided yet AET for such a long period and I can say that the existing AET data from the AMMA-CATCH observatory, for which I'm responsible, are full of gaps and will never look like it is in figure 9.

Reader COHARD noted that the comparison methodology is neither presented nor discussed. We have indicated in the article that reanalysis data are compared with data from point stations. Similarly, we have presented the distances between the reanalysis pixel under consideration and the station point in Table 1. This will be explained in more detail in the improved version.

With regard to flow, the reader is concerned to know the principle of comparison. In fact, at the point where the pixel is assumed to be an outlet, the cumulative head of water is collected over a particular period of time which depends on the data extracted. For the reanalysis, the cumulative period extends over 1 hour ending at the date and time of validity. Otherwise, the flow rate at the pixel point is expressed in m/h and not in meters as mentioned by the COHARD reader. All these details will be clarified in the new version, including how the daily Runoff data are returned to the same unit as that of the reanalyses. The reanalysis pixel should be seen as a known standard base (0.1° x 0.1° for LAND and 0.25° x 0.250 for ERA5) on which the height of water arriving each hour is measured.

The reader stated that the inflow from neighboring countries is never mentioned or discussed, as the authors focus solely on Benin. This assertion is not verified. In the discussion section, and in particular lines 368 to 372, the following is stated: "*Furthermore, the observations are in m³/hour, which has been broken down by the surface area of each basin in order to bring them back to the same unit with the reanalyses. The surface area considered in this operation is nothing other than the surface area of the hydrological basin and not the surface area of the hydrogeological basin, which is generally larger than the former*."

The given surface area considered in the calculation represents the surface area of the basin at the outlet where the assessment is made and not the surface area of any of the four basins. In summary, we considered the runoff from the watersheds at each outlet. Although these sub-basins are located in the four hydrological complexes, they do not have an independent relationship. This is the case for the basin of the river Ouémé at Bétérou, which drains less surface area than the basin of the river Ouémé at the Bonou outlet, all of which are located in the large hydrological unit known as Ouémé-YEWA.

The reader assumed that the in-situ piezometric level data were measured together with the underground runoff data. Depending on whether the water is recharging the water table or rising by capillary action, the underground runoff value may take on a negative or positive sign, which does not mean that the water is above or below the Earth's surface. This variable does not correspond to the piezometric level. Rather, it is the "*Lake mix-layer depth*", which corresponds to the depth at which we mix with "*lakes, reservoirs, rivers and coastal waters*". All these details will be given in the improved version to be submitted so that readers can refer to the appropriate data as needed. A photo of a piezometric measurement in the field (observation data used for comparison) is shown.

[Figure]

Piezometric level measurements in the field

Returning to the subject of flow, it should be pointed out that the in-situ measurements are taken at the outlet, i.e. the cumin point where the water in the basin is moved. At this point, the water arriving is not just surface water. It can also originate from Hortonian run-off, or from baseflow. In this respect, it is absurd to compare the data measured in situ at the outlets with the surface data as indicated by COHARD. The plots on the monthly cycle of surface flow are presented as follows:

[Figure]

Figure 1: Monthly cycle of surface flow for the two reanalyses.

Finally, the reader COHARD pointed out that PET and TEA seem to be confused. We take note of this information. We have no difficulty considering observational data such as PET rather than AET. It is a little surprising to observe a large phase shift between the observed PET data and PET from the reanalyses (Figure 2), but we trust the COHARD reader. He recently shared with fist author the remaining AET data at the Nalohou station (Figure 3). They are available until 2017 but we noted that there was too much missing data after 2014. We will certainly explain this in the revised manuscript.

[Figure]

Figure 2: Comparison of PET data at Nalohou and Béléfoungou station

[Figure]

Figure 3 : Monthly variability of ETR at Nalohou station

4) The evaluation only comments on metrics without describing time series. This makes the analysis often inconsistent.

This comment by COHARD contradicts his comment 2, which states that assertions are given without any criteria or metrics to justify them. Here, he pointed out that the metrics are commented upon without describing the time series. Nevertheless, we will take care to improve the document on the basis of his criticisms.

Thank you, COHARD, for your interest in the work and your comments.